# Impact of combined pulmonary fibrosis and emphysema on lung cancer risk and mortality in rheumatoid arthritis: A multicenter retrospective cohort study

**Shunsuke Mori** [1]*, **Yukitaka Ueki**[2], **Mizue Hasegawa**[3], **Kazuyoshi Nakamura**[4], **Kouya Nakashima**[5], **Toshihiko Hidaka**[6], **Koji Ishii**[7], **Hironori Kobayashi**[8], **Tomoya Miyamura**[9]

1 Department of Rheumatology, Clinical Research Center for Rheumatic Diseases, National Hospital Organization (NHO) Kumamoto Saishun Medical Center, Kohshi, Kumamoto, Japan, 2 Rheumatic and Collagen Disease Center, Sasebo Chuo Hospital, Sasebo, Nagasaki, Japan, 3 Department of Respiratory Medicine, Tokyo Women's Medical University Yachiyo Medical Center, Yachiyo, Chiba, Japan, 4 Department of Respiratory Medicine, NHO Kumamoto Saishun Medical Center, Kohshi, Kumamoto, Japan, 5 Department of Radiology, NHO Kumamoto Saishun Medical Center, Kohshi, Kumamoto, Japan, 6 Institute of Rheumatology, Miyazaki-Zenjinkai Hospital, Miyazaki, Japan, 7 Department of Rheumatology, Oita Red Cross Hospital, Oita, Japan, 8 Department of Thoracic Surgery, NHO Kumamoto Saishun Medical Center, Kohshi, Kumamoto, Japan, 9 Department of Internal Medicine and Rheumatology, Clinical Research Institute, NHO Kyushu Medical Center, Fukuoka, Japan

* mori.shunsuke.ra@mail.hosp.go.jp

**Data Availability Statement:** All data supporting the findings are available from the Human Research Ethics Committee of National Hospital

## Abstract

### Objective

Combined pulmonary fibrosis and emphysema (CPFE) is a syndrome characterized by the coexistence of emphysema and fibrotic interstitial lung disease (ILD). The aim of this study was to examine the effect of CPFE on lung cancer risk and lung cancer-related mortality in patients with rheumatoid arthritis (RA).

### Methods

We conducted a multicenter retrospective cohort study of patients newly diagnosed with lung cancer at five community hospitals between June 2006 and December 2021. Patients were followed until lung cancer-related death, other-cause death, loss to follow-up, or the end of the study. We used the cumulative incidence function with Gray's test and Fine-Gray regression analysis for survival analysis.

### Results

A total of 563 patients with biopsy-proven lung cancer were included (82 RA patients and 481 non-RA patients). The prevalence of CPFE was higher in RA patients than in non-RA patients (40.2% vs.10.0%) at lung cancer diagnosis. During follow-up, the crude incidence rate of lung cancer-related death was 0.29 and 0.10 per patient-year (PY) in RA and non-RA patients, and 0.32 and 0.07 per PY in patients with CPFE and patients without ILD or

Organization Kumamoto Saishun Medical Center for all interested researchers who meet the criteria for access to confidential data. Because these data include patients' personal information, the Committee does not recommend that such data be made public unnecessarily. Please contact Mr. Masahiro Hamaguchi, the Control Manager of the Committee, at 616-syol@mail.hosp.go.jp to request the data.

**Funding:** This study was supported by internal research funds from the National Hospital Organization (NHO), Japan, as well as unconditional research funds from AbbVie GK, Asahikasei Pharma Corp., and Chugai Pharmaceutical Co. Ltd. The funders had no role in the study design, data collection and analysis, decision to publish, or manuscript preparation. No additional external funding was received for this study.

**Competing interests:** I have read the journal's policy and the authors of this manuscript have the following competing interests: S. Mori received honoraria for lectures from AbbVie GK, Eli Lilly Japan K.K., Pfizer Japan Inc., Chugai Pharmaceutical Co. Ltd., Janssen Pharmaceutical K.K., Boehringer Ingelheim Japan, and Taisho Pharma Co., Ltd. and received research funds from AbbVie GK, Asahikasei Pharma Corp, and Chugai Pharmaceutical Co., Ltd. Y. Ueki received honoraria for lectures from AbbVie GK, Eli Lilly Japan K.K., Pfizer Japan Inc., Asahikasei Pharma Corp., Astellas Pharma Inc., Bristol-Myers K.K., Chugai Pharmaceutical Co. Ltd., Janssen Pharmaceutical K.K., Mitsubishi Tanabe Pharma Co., Ono Pharmaceutical Co., and Takeda Pharmaceutical Co., Ltd. T. Hidaka received honoraria for lectures from AbbVie GK, Eli Lilly Japan K.K., Pfizer Japan Inc., Asahi Kasei Pharma Corp., Bristol-Myers K.K., Chugai Pharmaceutical Co., Ltd., and Eisai Co. K. Nakamura received honoraria for lectures from AstraZeneca K.K. The other authors had no financial relationships that could create a potential conflict of interest or the appearance of a conflict of interest with regard to the work. This does not alter our adherence to PLOS ONE policies on sharing data and materials.

emphysema, respectively. The estimated death probability at 5 years differed between RA and non-RA patients (66% vs. 32%, *p*<0.001) and between patients with CPFE and patients without ILD or emphysema (71% vs. 24%, *p*<0.001). In addition to clinical cancer stage and no surgery within 1 month, RA and CPFE were identified as independent predictive factors for increased lung cancer-related mortality (RA: adjusted hazard ratio [HR], 2.49; 95% confidence interval [CI], 1.65–4.76; CPFE: adjusted HR 2.01; 95% CI 1.24–3.23).

## Conclusions

RA patients with lung cancer had a higher prevalence of CPFE and increased cancer-related mortality compared with non-RA patients. Close monitoring and optimal treatment strategies tailored to RA patients with CPFE are important to improve the poor prognosis of lung cancer.

## Introduction

Rheumatoid arthritis (RA) is a chronic immune-mediated inflammatory disease that is characterized by persistent synovitis and progressive damage to multiple joints [1, 2]. RA-associated systemic inflammation can also cause medical problems in multiple tissues and organs, such as interstitial lung disease (ILD), vasculitis, cardiovascular disease, and venous thromboembolism [3–6]. Multiple studies have shown a high prevalence of preclinical and clinical ILD throughout the RA disease course [5, 7]. In recent population-based matched cohort studies, RA patients with ILD had significantly increased mortality compared with those without ILD [8, 9], and this complication strongly contributed to excess mortality in RA compared with the general population [10, 11]. Although a variety of radiological and histological patterns of ILD are known to occur in RA patients, the most common pattern is usual interstitial pneumonia (UIP), following by nonspecific interstitial pneumonia (NSIP) [5, 12]. Similar to idiopathic pulmonary fibrosis (IPF, also known as idiopathic UIP), the UIP pattern in RA-associated ILD is associated with a worse prognosis compared with the NSIP pattern [5, 12, 13].

The presence of emphysema is relatively common in patients with fibrotic ILD, including IPF, and this has been designated "combined pulmonary fibrosis and emphysema" (CPFE) [14, 15]. In 2022, the research definition and classification criteria of CPFE as a clinical syndrome were formalized in an official statement by the American Thoracic Society (ATS), the European Respiratory Society (ERS), Japanese Respiratory Society (JRS), and the Latin American Thorax Association (ALAT) [15]. Epidemiological studies uncovered the association of lung cancer risk and IPF [16–18], and IPF seems to increase the risk of death in patients with lung cancer [19, 20]. An association of lung cancer and idiopathic CPFE has also been suggested. According to recent meta-analyses, the risk of lung cancer in patients with idiopathic CPFE was significantly higher than that in patients with IPF or emphysema alone, and survival in lung cancer patients with idiopathic CPFE was significantly shorter than in lung cancer patients without CPFE [21, 22].

Patients with RA continue to show an increased risk of lung cancer compared with the general population, and lung cancer is one of the most frequent malignancies in patients with RA [23–25]. Several risk factors for lung cancer in RA patients have been proposed, including male sex, smoking, and RA-associated ILD. In addition, chronic immune dysregulation and inflammation may contribute to neoplastic processes [26]. Cottin et al. first described the

CPFE syndrome in a series of patients with connective tissue disease (CTD), which prompted its recognition as a novel and distinct pulmonary manifestation within the spectrum of CTD-associated lung diseases, mainly occurring in current or former smokers with RA [27]. RA-associated CPFE is an entity different from RA-associated ILD [27, 28]. Nevertheless, few attempts have been made to identify a possible association of RA-associated CPFE and lung cancer.

To examine the impact of CPFE on lung cancer risk and lung cancer-related mortality in RA patients, we conducted a multicenter retrospective cohort study for patients who were newly given a biopsy-proven diagnosis of lung cancer in five participating institutions between June 2006 and December 2021. We examined the prevalence and radiological characteristics of CPFE in these patients. Additionally, we estimated cumulative incidence rates of lung cancer-related death over time and compared mortality estimates between RA and non-RA patients as well as between patients with CPFE and those without ILD or emphysema. Using Fine-Gray competing risks regression analysis, we calculated hazard ratios (HRs) of RA and CPFE for lung cancer-related death after adjusting for various baseline covariates.

## Patients and methods

### Patients

Using real-world databases of participating institutions, we identified RA patients who were newly given a biopsy-proven diagnosis of lung cancer in the rheumatology divisions of the following five community hospitals in Japan between June 2006 and December 2021: National Hospital Organization (NHO) Kumamoto Saishun Medical Center, Sasebo Chuo Hospital, Miyazaki Zenjinkai Hospital, Oita Red Cross Hospital, and NHO Kyushu Medical Center. All patients were required to fulfill the 1987 American College of Rheumatology (ACR) criteria or the 2010 ACR/European League Against Rheumatism (EULAR) criteria for diagnosis of RA [29, 30]. As non-RA controls, we identified patients without any type of immune-mediated inflammatory diseases who were newly diagnosed with biopsy-proven lung cancer in the respiratory disease division of NHO Kumamoto Saishun Medical Center during the same period as the enrollment of RA patients. Participants in this study were required to be 18 years of age or older. Patients who had previous history of cancer and those who were diagnosed with metastatic lung cancer were excluded from this study.

### Study design

We reviewed computer-based electronic patient records to scrutinize clinical data at the time of lung cancer diagnosis and during follow-up. These data were extracted from each institution's database by site investigators, and data sheets were submitted to the data center for this study at the NHO Kumamoto Saishun Medical Center. To reduce inter-center or inter-physician differences, the data center reviewed all data submitted by the site investigators, especially focusing on fulfillment of the inclusion and exclusion criteria at enrollment, data quality at baseline, and the cause of death during follow-up. Questions were resolved by consensus of the investigator team.

Clinical characteristics at baseline were obtained when the diagnosis of lung cancer was made, which included demographic characteristics (age and sex), RA-related data (disease duration, RA disease activity, radiological Steinbrocker stage, anticyclic citrullinated peptide antibody [anti-CCP], rheumatoid factor [RF], and the use of disease-modifying antirheumatic drugs [DMARDs]), and smoking history. Desaturation with exercise, which was defined as percutaneous oxygen saturation (SpO$_2$) <90% during a 6-minute walk, was also measured for RA patients. Histological classification and clinical staging of lung cancer at diagnosis, as well

as initial treatment of lung cancer, were also obtained as baseline characteristics. For diagnosis of ILD, emphysema, and CPFE, we reviewed high-resolution computed tomography (HRCT) images taken at the diagnosis of lung cancer.

Follow-up started on the day of biopsy-proven diagnosis of lung cancer and ended with lung cancer-related death, other-cause death, loss to follow-up, or the last follow-up visit before July 31, 2022, whichever was first. Patients who missed two or more scheduled visits without any contact were classified as lost to follow-up.

## Histological classification, clinical staging, and treatment of lung cancer

Lung cancer was classified into three histological types (adenocarcinoma, squamous cell carcinoma, and small cell carcinoma) by lung biopsy (morphology and immunohistochemistry) [31]. In this cohort, no patients had a diagnosis of large cell carcinoma. The clinical staging of lung cancer was determined according to the tumor, node, metastasis (TNM) classification system [32, 33]. Initial treatments of lung cancer, including surgery, chemotherapy, and radiation therapy, were examined.

## HRCT pattern of ILD and diagnosis of CPFE

HRCT images at the time of lung cancer diagnosis were viewed in random order and independently by three observers (a board-certified radiologist [K. Nakashima] and two board-certified pulmonologists [M. Hasegawa and K. Nakamura]) who were blinded to the patients' clinical status. Final decisions were made by consensus if there were disagreements.

The diagnosis and classification of ILD was made according to the updated official ATS/ERS/JRS/ALAT clinical practice guideline [34, 35]. Each patient with ILD was classified as having one of the following four HRCT patterns: definite UIP pattern, probable UIP pattern, indeterminate UIP pattern, and alternative diagnosis. The definition of CPFE was according to the 2022 official ATS/ERS/JRS/ALAT research statement as follows: CPFE is defined as coexistence of both pulmonary fibrosis (regardless of the type of fibrotic ILD) and emphysema on HRCT. Emphysema was defined as well-demarcated areas of low attenuation delimitated by a very thin wall ($\leq$ 1 mm) or no wall and involving at least 5% of total lung volume [15].

## Cause of death

The cause of death was determined according to each treating physician's judgment. Lung cancer-related death was defined as death caused by cachexia, organ failure, infectious disease (such as pneumonia and sepsis), complications of cancer therapies, or oncological emergencies.

## Ethics approval

This study was conducted in accordance with the principles of the Declaration of Helsinki. The protocol of this study met the requirements of the Ethical Guidelines for Medical and Health Research Involving Human Subjects, Japan, and was approved by the Human Research Ethics Committee at NHO Kumamoto Saishun Medical Center (No. 2–21), the Institutional Review Boards of Sasebo Chuo Hospital (No. 2020–30), Oita Red Cross Hospital (No. 257), Miyazaki Zenjinkai Hospital (No. 20201028), and the Human Research Ethics Committee at NHO Kyushu Medical Center (No. 21C035). Because the study involved a retrospective review of patient records and the data were analyzed anonymously, our ethics committees waived the requirement for patient informed consent to participate.

### Sample size

To estimate the sample size in a survival study for time-to-event, the expected probability of lung cancer-related death or event-free survival at the time point of interest is needed. However, it is difficult to determine the expected probability in patients with or without RA-associated CPFE due to the lack of previous data. We were therefore unable to estimate sample size for this study. The number of patients newly diagnosed with lung cancer in the participating institutions during the enrollment period determined the sample size.

### Statistical analysis

Regarding clinical characteristics at lung cancer diagnosis, mean and standard deviation (SD) were used as descriptive statistics for data with a continuous distribution, which included non-normally distributed data [36], and number (percentage) was used for categorical data. To compare baseline characteristics between the RA and non-RA groups, we performed Fisher's exact probability test for categorial variables and the independent-measures *t*-test for continuous variables. No participants had missing data regarding the baseline characteristics that we planned to use in analyses.

Crude incidence rates of lung cancer-related death and 95% confidence intervals (CIs) were calculated by dividing the number of incidence cases by the number of corresponding follow-up patient-years (PYs).

Cumulative incidence of lung cancer-related death, defined as the probability that a lung cancer-related death event has occurred before a given time, was estimated using the cumulative incidence function (CIF), because we considered the presence of competing risks (namely, loss to follow-up and other-cause death). Gray's test with or without the *post hoc* Holm's procedure was used to test the equality of CIF plots among patient groups [37].

Fine-Gray competing risks regression analysis was performed to evaluate the effect of each of the baseline characteristics on lung cancer-related death outcome over time and to calculate adjusted HRs with 95% CIs. As predictor variables, we used a set of baseline characteristics that were considered to be clinically relevant and important based on previous knowledge, which included demographic data, RA or non-RA, smoking history, HRCT-based CPFE diagnosis, histological type of lung cancer, TNM clinical stage, and surgery performed within 1 month of diagnosis. For the RA group, RA disease duration, RA disease activity, and Steinbrocker radiological stage as a measure of RA progression at baseline were also included in the analysis as predictor variables. The use of DMARDs at lung cancer diagnosis was not included in Fine-Gray regression analysis because these drugs were discontinued during cancer treatment [38]. Predictor variables with *p*-values <0.10 in univariable models were used in a multivariable Fine-Gray regression analysis. A forced entry procedure was used in the multivariable model.

Two-sided *p*-values <0.05 were considered to indicate statistical significance. All calculations were performed using PASW Statistics version 27 (SPSS Japan Inc., Tokyo, Japan) and Easy R (Saitama Medical Center, Jichi Medical University, Saitama, Japan) [39].

## Results

### Clinical characteristics at lung cancer diagnosis

A total of 563 patients (82 RA and 481 non-RA patients) were included in the present study. Clinical characteristics at the time of lung cancer diagnosis are shown in Table 1. No patients had missing data regarding baseline characteristics. In RA patients, the mean duration of RA was 11.4 years, and approximately 70% of patients had an RA duration of >5 years. RA disease

**Table 1. Clinical characteristics at diagnosis of lung cancer.**

| | RA patients | Non-RA patients[†] | p-value* |
|---|---|---|---|
| | (n = 82) | (n = 481) | |
| Age, years, mean (SD) | 71.4 (8.2) | 72.0 (9.0) | 0.59 |
| Male, number (%) | 44 (53.7) | 309 (64.2) | 0.083 |
| RA duration, years, mean (SD) | 11.4 (10.5) | – | – |
| ≤5 years, number (%) | 24 (29.3) | – | – |
| RA activity, low or remission, number (%) | 56 (68.3) | – | – |
| Steinbrocker stages III/IV, number (%) | 31 (37.8) | – | – |
| Anti-CCP or RF positive, number (%) | 81 (98.8) | – | – |
| SpO$_2$ <90% during a 6-minute walk, number (%) | 39 (47.6) | – | – |
| Smoking history | | | |
| ≥30 PYs, number (%) | 40 (48.8) | 246 (51.1) | 0.72 |
| Never-smoker, number (%) | 24 (29.3) | 159 (33.1) | 0.53 |
| HRCT-based CPFE diagnosis, number (%) | | | |
| CPFE | 33 (40.2) | 48 (10.0) | <0.001 |
| Definite UIP pattern | 4/33 (12.1) | 8/48 (16.7) | 0.75 |
| Probable UIP pattern | 0 | 2/48 (4.2) | 0.51 |
| Indeterminate UIP pattern | 24/33 (72.7) | 30/48 (62.5) | 0.47 |
| Alternative diagnosis[‡] | 5/33 (15.2) | 8/48 (16.7) | 1.00 |
| ILD alone | 8 (9.8) | 7 (1.5) | <0.001 |
| Definite UIP pattern | 3/8 (37.5) | 2/7 (28.6) | 1.00 |
| Probable UIP pattern | 2/8 (25) | 2/7 (28.6) | 1.00 |
| Indeterminate UIP pattern | 1/8 (12.5) | 0 | 1.00 |
| Alternative diagnosis[‡] | 2/8 (25) | 3/7 (42.9) | 0.61 |
| Emphysema alone | 19 (23.2) | 128 (26.6) | 0.59 |
| Without ILD or emphysema | 22 (26.8) | 298 (62.0) | <0.001 |
| Histological type of lung cancer, number (%) | | | |
| Overall patients | | | |
| Adenocarcinoma | 43 (52.4) | 342 (71.1) | <0.001 |
| Squamous cell carcinoma | 27 (32.9) | 76 (15.8) | <0.001 |
| Small cell carcinoma | 12 (14.6) | 63 (13.1) | 0.73 |
| Patients with CPFE | | | |
| Adenocarcinoma | 13/33 (39.4) | 19/48 (39.6) | 1.00 |
| Squamous cell carcinoma | 13/33 (39.4) | 15/48 (31.3) | 0.48 |
| Small cell carcinoma | 7/33 (21.2) | 14/48 (29.2) | 0.45 |
| TNM clinical stage of lung cancer, number (%) | | | |
| Overall patients | | | |
| Stage I | 34 (41.5) | 221 (45.9) | 0.47 |
| Stage II | 12 (14.6) | 55 (11.4) | 0.46 |
| Stage III | 19 (23.2) | 77 (16.0) | 0.11 |
| Stage IV | 17 (20.7) | 128 (26.6) | 0.34 |
| Patients with CPFE | | | |
| Stage I | 10/33 (30.3) | 11/48 (22.9) | 0.61 |
| Stage II | 6/33 (18.2) | 10/48 (20.8) | 1.00 |
| Stage III | 9/33 (27.3) | 11/48 (22.9) | 0.79 |
| Stage IV | 8/33 (24.2) | 16/48 (33.3) | 0.46 |
| Treatment of lung cancer,[§] number (%) | | | |
| Surgery | 46 (56.1) | 286 (59.5) | 0.55 |

(*Continued*)

**Table 1.** (Continued)

| | RA patients | Non-RA patients[†] | p-value* |
|---|---|---|---|
| | **(n = 82)** | **(n = 481)** | |
| Chemotherapy and/or radiation therapy | 21 (25.6) | 162 (33.7) | 0.16 |
| No treatment | 15 (18.3) | 33 (6.9) | 0.002 |

*Compared between RA patients and non-RA patients using Fisher's exact probability test for categorical variables and independent-measures *t*-test for continuous variables.

[†]Non-RA patients did not have a diagnosis of any type of immune-mediated inflammatory disease.

[‡]All cases were diagnosed as fibrotic NSIP.

[§]Defined as the initial treatment within 1 month after the diagnosis of lung cancer.

RA, rheumatoid arthritis; anti-CCP, anti-cyclic citrullinated peptide antibodies; RF, rheumatoid factor; $SpO_2$, oxygen saturation as measured using pulse oximeter; PYs, pack-years; HRCT, high-resolution computed tomography; CPFE, combined pulmonary fibrosis and emphysema; ILD, interstitial lung disease; UIP, usual interstitial pneumonia; NSIP, non-specific interstitial pneumonia; TNM, tumor, node, and metastasis; SD, standard deviation.

activity was well controlled in 68.3% of patients, but desaturation with exercise ($SpO_2 <90\%$ during a 6-minute walk) was observed in 47.6% of RA patients. More than 90% of RA patients were receiving DMARD therapy and half were receiving methotrexate (MTX) as monotherapy or in combination with a biological DMARD (bDMARD) or other conventional synthetic DMARD (csDMARD) at the time of lung cancer diagnosis (S1 Table).

Smoking history rates were similar between RA and non-RA patients (Table 1). The rate of patients with CPFE was higher in RA patients compared with non-RA patients (40.2% vs. 10.0%). The predominant HRCT pattern of CPFE was indeterminant UIP in RA patients (72.7%) and in non-RA patients (62.5%), and the distribution of HRCT patterns was not significantly different between the patient groups. All patients having HRCT findings suggestive of an alternative diagnosis were diagnosed with fibrotic NSIP. Among CPFE patients, 2 had been diagnosed with pulmonary hypertension using echocardiography and right heart catheterization (1 patient with RA and 1 patient without RA).

For the RA group, there was no significant difference in RA duration, RA disease activity, or RA progression (Steinbrocker radiological stage) among patients with CPFE, those with ILD or emphysema alone, and those without these complications (S2 Table). In contrast, the prevalence of patients with desaturation with exercise ($SpO_2 <90\%$) was significantly different among these three patient groups ($p <0.001$). Typical HRCT images of CPFE associated with RA are presented in S1 Fig.

The predominant histological type of lung cancer was adenocarcinoma in both RA and non-RA groups, followed by squamous cell carcinoma and small cell carcinoma. However, the rate of adenocarcinoma was significantly lower in RA patients than in non-RA patients (52.4% vs. 71.1%), and squamous cell carcinoma was more often observed in RA patients than non-RA patients (32.9% vs.15.8%). Such differences were not observed when considering just patients with CPFE (Table 1).

Concerning TNM clinical stage, more than 40% of patients were at stage I at lung cancer diagnosis in the RA and non-RA groups, and there were no significant differences in the frequency of each stage between patient groups (Table 1). Almost all patients at TNM stages I and II underwent surgery within 1 month after the cancer diagnosis. The rate of patients undergoing surgery was similar in both patient groups (56.1% vs. 59.5%). Chemotherapy or radiation therapy was not introduced for the majority of patients with pulmonary fibrosis because of the increased risk of disease exacerbation.

**Table 2. Mortality in patients newly diagnosed with lung cancer.**

| | All patients (n = 563) | RA patients (n = 82) | Non-RA patients (n = 481) |
|---|---|---|---|
| Follow-up,* years, mean (95% CI) | 2.9 (2.7–3.1) | 2.0 (1.5–2.5) | 3.1 (2.8–3.3) |
| Lost to follow-up, number (%) | 149 (26.5) | 19 (23.2) | 130 (27.0) |
| Other-cause death,[†] number (%) | 5 (0.9) | 0 | 5 (1.0) |
| Lung cancer-related death, number (%) | 187 (33.2) | 47 (57.3) | 140 (29.1) |
| Crude incidence rate per PY (95% CI) | 0.12 (0.10–0.14) | 0.29 (0.22–0.38) | 0.10 (0.08–0.11) |
| CIF mortality estimates[‡] | | | |
| Time to death, years, median (95% CI) [§] | ND | 2.5 (1.8–4.8) | ND |
| Cumulative incidence at 5 years (95% CI)[††] | 0.36 (0.32–0.41) | 0.66 (0.52–0.76) | 0.32 (0.27–0.36) |

*Follow-up was measured from the diagnosis of lung cancer.

[†]Included cardiac infarction, aortic aneurysm rupture, acute subdural hemorrhage, liver failure due to primary hepatocellular carcinoma, and acute bleeding from primary gastric cancer.

[‡]Gray's test was used for comparisons of CIF mortality estimates over time between RA patients and non-RA patients ($p < 0.001$).

[§]Median time to lung-cancer related death was the estimated time at which 50% of patients would have died due to lung cancer.

[††]Cumulative incidences of lung cancer-related death at 5 years were the estimated 5-year mortality rates.

RA, rheumatoid arthritis; PY, patient-year; CIF, cumulative incidence function; CI, confidence interval; ND, not determined.

## Mortality in RA and non-RA patients with lung cancer

After the diagnosis of lung cancer was made, patients were followed for a mean of 2.9 years (95% CI 2.7–3.1). As shown in Table 2, 149 patients (26.5%) were lost to follow-up. Lung cancer-related death occurred in 187 patients (47 cases in the RA group [57.3%] and 140 cases in the non-RA group [29.1%]). Among these patients, 9 developed respiratory failure due to an acute exacerbation of ILD or CPFE (6 in RA patients and 3 in non-RA patients), and 36 died from infectious disease (pneumonia and sepsis; 16 in RA patients and 20 in non-RA patients). The 2 patients with CPFE and pulmonary hypertension died from oncological emergency events. Death from other causes occurred in only 5 non-RA patients. The crude incidence rate of lung cancer-related death was higher in RA patients vs. non-RA patients (0.29 per PY [95% CI 0.22–0.38] for RA patients and 0.10 per PY [95% CI 0.08–0.11] for non-RA patients).

According to the CIF, the median time to lung cancer-related death was 2.5 years (95% CI 1.98–4.8) in RA patients. The cumulative incidence at 5 years was estimated to be 0.66 (95% CI 0.52–0.76) for RA patients and 0.32 (95% CI 0.27–0.36) for non-RA patients (Table 2). Significant differences in CIF mortality estimates over time were observed between both patient groups ($p < 0.001$ with Gray's test). CIF plots of lung cancer-related death in RA and non-RA patients are shown in Fig 1.

## Mortality in lung cancer patients grouped according to the presence of CPFE

As shown in Table 3, lung cancer-related death occurred in 50 patients (61.7%) in the CPFE group, 65 (40.1%) in the ILD or emphysema alone group, and 72 (22.5%) in the group without ILD or emphysema. The crude incidence rate of lung cancer-related death was highest in the CPFE group (0.32 per PY [95% CI 0.24–0.42] for CPFE patients, 0.17 per PY [95% CI 0.13–0.21] for patients with ILD or emphysema alone, and 0.07 per PY [95% CI 0.05–0.08] for patients without ILD or emphysema). Such differences were observed separately in RA and non-RA patients (among RA patients: 0.50 per PY [95% CI 0.33–0.74] for CPFE patients, 0.22 per PY [95% CI 0.13–0.37] for patients with ILD or emphysema alone, and 0.18 per PY [95% CI 0.10–0.34] for patients without ILD or emphysema; among non-RA patients: 0.24 per PY [95% CI 0.16–0.35] for CPFE patients, 0.16 per PY [95% CI 0.12–0.21] for patients with ILD or

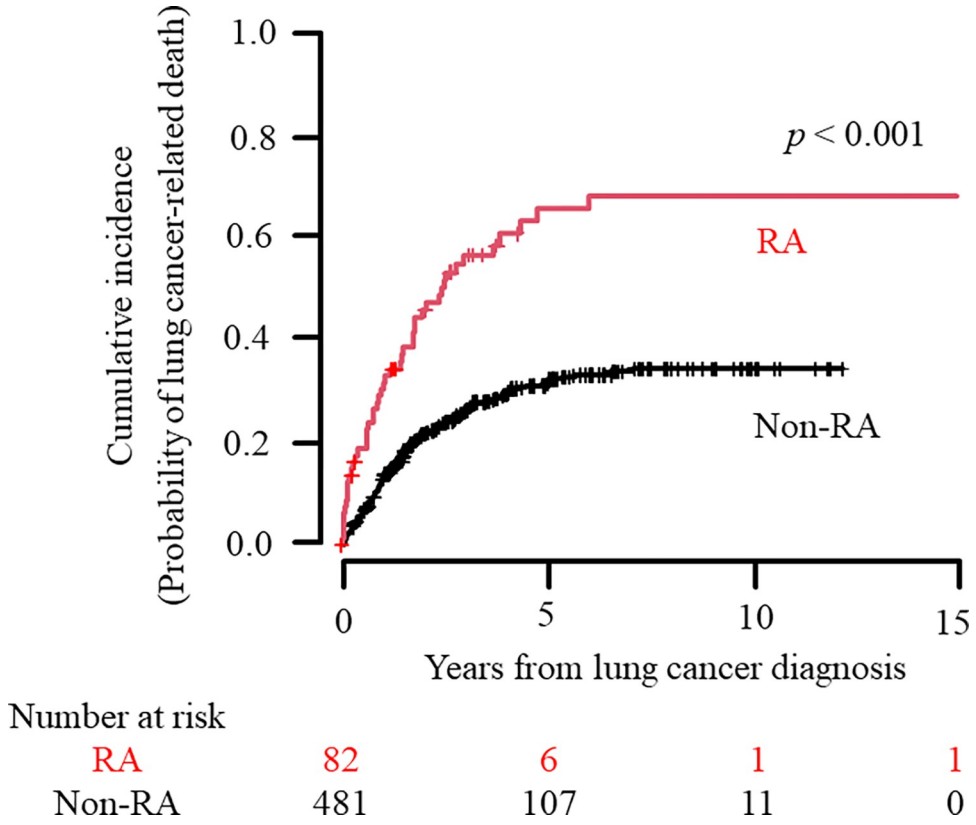

**Fig 1. Cumulative incidence of lung cancer-related death in RA and non-RA patients.** Using the CIF, the cumulative incidence of lung cancer-related death in patients who were newly given a diagnosis of lung cancer is shown for the RA and non-RA groups. Numbers below these figures represent the number of patients at risk. The cumulative incidence of death over time between both groups was compared using Gray's test ($p < 0.001$). RA, rheumatoid arthritis; CIF, cumulative incidence function.

emphysema alone, and 0.06 per PY [95% CI 0.05–0.08] for patients without ILD or emphysema; Table 3).

According to the CIF, the median time to lung cancer-related death was 2.1 years (95% CI 1.5–3.0) in the CPFE group. The cumulative incidence of lung cancer-related death at 5 years was estimated to be 0.71 (95% CI 0.58–0.81) for the CPFE group, 0.48 (95% CI 0.39–0.57) for the ILD or emphysema alone group, and 0.24 (95% CI 0.19–0.30) for the group without ILD or emphysema (Table 3). Significant differences in CIF mortality estimates over time were observed among the three groups ($p < 0.001$ with Gray's test). Such differences were observed separately in RA and non-RA patients ($p = 0.006$ for a comparison of the three groups in RA patients and $p < 0.001$ for a comparison of the three groups in non-RA patients with Gray's test; Table 3). Additionally, RA patients were estimated to be at higher risk of lung cancer-related death over time compared with non-RA patients in all 3 groups ($p < 0.001$ with Gray's test; Table 3).

CIF plots for lung cancer-related death in the CPFE group, the ILD or emphysema alone group, and the group without ILD or emphysema are shown in Fig 2.

### Predictive factors for lung cancer-related death in patients newly diagnosed with lung cancer

Results of univariable and multivariable Fine-Gray competing risks regression analyses are shown in Table 4. All predictor variables with $p$-values < 0.10 in univariable analyses were

**Table 3. Mortality in patients who were newly diagnosed with lung cancer grouped according to the HRCT-based CPFE diagnosis.**

| | CPFE | ILD or emphysema alone | Without ILD or emphysema |
|---|---|---|---|
| All patients (n = 563) | (n = 81) | (n = 162) | (n = 320) |
| Follow-up,* years, mean (95% CI) | 1.9 (1.5–2.4) | 2.4 (2.1–2.7) | 3.4 (3.1–3.7) |
| Lost to follow-up, number (%) | 14 (17.3) | 38 (23.5) | 97 (30.3) |
| Other-cause death, number (%) | 1 (1.2) | 3 (1.9) | 1 (0.3) |
| Lung cancer-related death, number (%) | 50 (61.7) | 65 (40.1) | 72 (22.5) |
| Crude incidence rate per PY (95% CI) | 0.32 (0.24–0.42) | 0.17 (0.13–0.21) | 0.07 (0.05–0.08) |
| Time to death, years, median (95% CI)[†] | 2.1 (1.5–3.0) | 6.1 (3.9–ND) | ND |
| Cumulative incidence at 5 years (95% CI)[†] | 0.71 (0.58–0.81) | 0.48 (0.39–0.57) | 0.24 (0.19–0.30) |
| RA patients (n = 82) | (n = 33) | (n = 27) | (n = 22) |
| Follow-up,* years, mean (95% CI) | 1.5 (0.7–2.2) | 2.2 (1.4–3.1) | 2.5 (1.1–3.9) |
| Lost to follow-up, number (%) | 5 (15.2) | 6 (22.2) | 8 (36.4) |
| Other-cause death, number (%) | 0 | 0 | 0 |
| Lung cancer-related death, number (%) | 24 (72.7) | 13 (48.1) | 10 (45.5) |
| Crude incidence rate per PY (95% CI) | 0.50 (0.33–0.74) | 0.22 (0.13–0.37) | 0.18 (0.10–0.34) |
| Time to death, years, median (95% CI)[†] | 1.1 (0.6–2.5) | 3.7 (1.8–ND) | 4.8 (2.4–ND) |
| Cumulative incidence at 5 years (95% CI)[†] | 0.82 (0.63–0.92) | 0.68 (0.40–0.85) | 0.52 (0.28–0.71) |
| Non-RA patients (n = 481) | (n = 48) | (n = 135) | (n = 298) |
| Follow-up,* years, mean (95% CI) | 2.2 (1.6–2.8) | 2.4 (2.0–2.8) | 3.5 (3.1–3.8) |
| Lost to follow-up, number (%) | 9 (18.8) | 32 (23.7) | 89 (29.9) |
| Other-cause death, number (%) | 1 (2.1) | 3 (2.2) | 1 (0.3) |
| Lung cancer-related death, number (%) | 26 (54.2) | 52 (38.5) | 62 (20.8) |
| Crude incidence rate per PY (95% CI) | 0.24 (0.16–0.35) | 0.16 (0.12–0.21) | 0.06 (0.05–0.08) |
| Time to death, years, median (95% CI)[†] | 3.0 (2.1–ND) | ND | ND |
| Cumulative incidence at 5 years (95% CI)[†] | 0.65 (0.47–0.79) | 0.45 (0.35–0.54) | 0.22 (0.17–0.27) |

*Follow-up was measured from the diagnosis of lung cancer.

[†]Median time to death and cumulative incidence at 5 years were estimated by CIF. Gray's test with the *post hoc* Holm's procedure was used for comparisons of CIF mortality estimates over time among the three patient groups. For all patients, the *p*-values are as follows: $p < 0.001$ for a comparison among the three groups, CPFE vs. without ILD or emphysema, and ILD or emphysema alone vs. without ILD or emphysema. For RA patients, the *p*-values are as follows: $p = 0.006$ for a comparison among the three groups; $p = 0.015$ for CPFE vs. without ILD or emphysema; $p = 0.082$ for ILD or emphysema alone vs. without ILD or emphysema. For non-RA patients, the *p*-values are as follows: $p < 0.001$ for a comparison among the three groups, CPFE vs. without ILD or emphysema, and ILD or emphysema alone vs. without ILD or emphysema. Gray's test was used for comparisons of CIF mortality estimates over time between RA patients and non-RA patients in each of the three patient groups ($p < 0.001$ for all patient groups).

RA, rheumatoid arthritis; CPFE, combined pulmonary fibrosis and emphysema; ILD, interstitial lung disease; PY, patient-year; CIF, cumulative incidence function; 95% CI, 95% confidence interval; ND, not determined.

included in the multivariable analysis using a forced entry procedure. Through multivariate modeling, RA (adjusted HR 2.49 [95% CI 1.65–4.76] vs. non-RA), concomitant CPFE (adjusted HR 2.01 [95% CI 1.24–3.23] vs. without ILD or emphysema), concomitant ILD or emphysema alone (adjusted HR 1.56 [95% CI 1.03–2.36] vs. without ILD or emphysema), and TNM clinical stage (adjusted HR 3.76 [95% CI 2.26–6.23] for stage II, adjusted HR 2.81 [95% CI 1.65–4.79] for stage III, and adjusted HR 4.82 [95% CI 2.81–8.27] for stage IV vs. stage I) were identified as independent predictive factors for lung cancer-related death. In addition, surgery performed within 1 month after cancer diagnosis was an independent factor predicting decreased mortality of lung cancer (adjusted HR 0.40 [95% CI 0.26–0.63] vs. no surgery within 1 month).

Considering only RA patients, concomitant CPFE (adjusted HR 2.14 [95% CI 1.06–4.31] vs. without ILD or emphysema), TNM clinical stage (adjusted HR 2.58 [95% CI 1.02–6.50] for

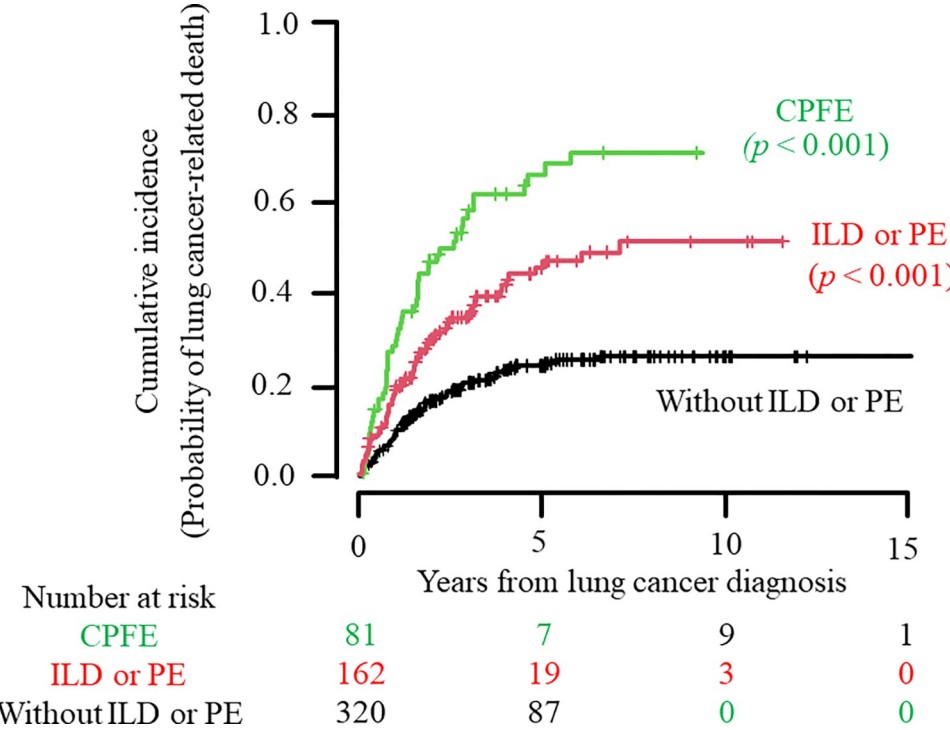

**Fig 2. Cumulative incidence of lung cancer-related death in all patients grouped by HRCT-based CPFE diagnosis.**
Using the CIF, the cumulative incidence of lung cancer-related death in patients who were newly given a diagnosis of lung cancer is shown in the CPFE group, the ILD or emphysema alone group, and the group without ILD or emphysema. Numbers below these figures represent the number of patients at risk. The cumulative incidence of death over time among groups was compared using Gray's test with the *post hoc* Holm's procedure ($p < 0.001$ for a comparison among the three groups, CPFE vs. without ILD or emphysema, and ILD or emphysema alone vs. without ILD or emphysema). HRCT, high-resolution computed tomography; CPFE, combined pulmonary fibrosis and emphysema; ILD, interstitial pneumonia; PE, pulmonary emphysema; CIF, cumulative incidence function.

stage II vs. stage I), and surgery within 1 month (adjusted HR 0.32 [95% CI 0.14–0.71] vs. no surgery) were independent predictive factors for lung cancer-related mortality. Neither RA duration, RA disease activity, nor Steinbrocker radiological stage was identified as a factor associated with lung cancer-related mortality (S3 Table). In a multivariable model in which desaturation with exercise (SpO$_2$ <90%) was used as a predictor variable instead of HRCT-based CPFE diagnosis, desaturation with exercise (adjusted HR 2.58 [95% CI 1.40–4.77]), TNM clinical stage (adjusted HR 2.68 [95% CI 1.16–6.21] for stage II vs. stage I), and surgery within 1 month (adjusted HR 0.37 [95% CI 0.17–0.84] vs. no surgery) were independent predictive factors for lung cancer-related mortality (S4 Table).

## Discussion

In this multicenter cohort study, CPFE was more frequently observed in RA patients than in non-RA patients when they were newly given a lung cancer diagnosis. The crude incidence rate of lung cancer-related death was higher in RA patients than in non-RA patients, and mortality estimates over time were significantly worse in RA patients compared with non-RA patients. Additionally, the crude incidence rate and mortality estimates of lung cancer-related death were significantly different among the CPFE group, the ILD or emphysema alone group, and the group without ILD or emphysema. Such differences were observed separately in RA and non-RA patients. Multivariable Fine-Gray competing risks regression analysis showed

**Table 4. Predictive factors for lung cancer-related death in all patients newly diagnosed with lung cancer.**

| | Unadjusted HR (95% CI) | *p*-value | Adjusted HR (95% CI) | *p*-value |
|---|---|---|---|---|
| Age per additional year | 1.00 (0.99–1.02) | 0.71 | – | – |
| Male vs. female | 1.98 (1.42–2.75) | <0.001 | 1.25 (0.80–1.95) | 0.32 |
| RA vs. non-RA | 2.64 (1.90–3.66) | <0.001 | 2.49 (1.65–4.79) | <0.001 |
| Smoking history ≥30 PYs | 2.27 (1.67–3.07) | <0.001 | 0.98 (0.62–1.56) | 0.94 |
| HRCT-based CPFE diagnosis | | | | |
| CPFE | 3.98 (2.77–5.70) | <0.001 | 2.01 (1.24–3.23) | 0.004 |
| ILD or emphysema alone | 2.09 (1.50–2.92) | <0.001 | 1.56 (1.03–2.36) | 0.037 |
| Without ILD or emphysema | 1 (reference) | – | 1 (reference) | – |
| Histological type of lung cancer | | | | |
| Adenocarcinoma | 1 (reference) | – | 1 (reference) | – |
| Squamous cell carcinoma | 2.41 (1.71–3.40) | <0.001 | 1.29 (0.86–1.94) | 0.22 |
| Small cell carcinoma | 2.86 (1.98–4.12) | <0.001 | 1.01 (0.66–1.55) | 0.96 |
| TNM clinical stage of lung cancer | | | | |
| Stage I | 1 (reference) | – | 1 (reference) | – |
| Stage II | 4.51 (2.68–7.58) | <0.001 | 3.76 (2.26–6.23) | <0.001 |
| Stage III | 5.72 (3.53–9.28) | <0.001 | 2.81 (1.65–4.79) | <0.001 |
| Stage IV | 10.38 (6.82–15.81) | <0.001 | 4.82 (2.81–8.27) | <0.001 |
| Surgery for lung cancer[†] | | | | |
| Yes vs. no | 0.17 (0.13–0.24) | <0.001 | 0.40 (0.26–0.63) | <0.001 |

Univariable and multivariable Fine–Gray competing risks regression analyses were conducted to evaluate baseline patient characteristics that predict lung cancer-related death outcome after lung cancer diagnosis. All variables with *p*-values < 0.10 in the univariable models were introduced into multivariable analysis using a forced entry procedure.

[†]Defined as surgery that was performed as the initial treatment within 1 month after the diagnosis of lung cancer.

RA, rheumatoid arthritis; PY, pack-year; HRCT, high-resolution computed tomography; CPFE, combined pulmonary fibrosis and emphysema, ILD, interstitial lung disease; TNM, tumor, node, and metastasis; HR, hazard ratio; 95% CI, 95% confidence interval.

that RA, CPFE, TNM stage, and no surgery within 1 month were independent predictive factors for lung cancer-related death.

Despite great advances in pharmacotherapy in the last two decades, malignancy still contributes to excess RA-related mortality compared with the general population [10, 40, 41]. In a national, matched cohort study with the United States (US) Veterans Health Administration database from 2000 to 2017, Johnson et al. showed that approximately 15% of excess deaths in RA were attributed to lung cancer (adjusted HR 1.47 [95% CI 1.37–1.57] vs. non-RA patients) [10]. In a retrospective cohort study with the Surveillance, Epidemiology, and End Results-Medicare database in the US from 1993 to 2013, Wang et al. showed that RA was associated with increased cancer-specific mortality among elderly patients with lung cancer (adjusted HR 1.13 [95% CI 1.09–1.16]) [42]. The present study was conducted in patients who developed lung cancer during 2006–2021, and the risk of lung cancer-related death was significantly higher in RA patients compared with non-RA patients (adjusted HR 2.49 [95% CI 1.65–4.79]). Chronic systemic inflammation and altered immunity in RA may affect the mortality associated with lung cancer [43]. In the present study, however, RA disease activity was not identified as a predictive factor for increased lung cancer-related mortality in RA patients. Nayak et al., using the Texas Cancer Registry and Medicare-linked database between 2001 and 2010, showed no increased mortality in elderly RA patients who developed lung cancer compared with non-RA patients [44]. In a single-center cohort study comparing overall survival between

lung cancer patients with autoimmune diseases (including RA) and those without autoimmune diseases in the US from 2003 to 2019, Jacob et al. showed that no individual type of autoimmune disease was associated with worse prognosis compared with controls, despite the fact that fewer patients in the autoimmune cohort received standard-of-care lung cancer treatment [45]. The reason for the conflicting findings regarding the association of RA with lung cancer-related mortality remains unclear. Rates of CPFE complication were not determined in previous studies.

In the present study, the adjusted HR (95% CI) of lung cancer-related mortality death was 2.01 (1.23–3.23) for CPFE patients compared with patients without ILD or emphysema. Considering RA patients alone, the adjusted HR (95% CI) for CPFE was 2.14 (1.06–4.31). Thus, concomitant CPFE significantly contributed to the increased lung cancer-related deaths in RA and non-RA patients. Recent meta-analyses regarding idiopathic CPFE and lung cancer showed that there was a significantly higher risk for lung cancer in patients with CPFE than in patients with IPF or emphysema alone [21, 22]. The median survival time was significantly shorter in lung cancer patients with concomitant idiopathic CPFE compared with those without CPFE (19.5 vs. 53.1 months) [21]. Regarding RA patients, data on the impact of CPFE on lung cancer-related mortality are lacking. In a single-center cohort study for RA patients in Japan from 2009 to 2014, Kakutani et al. showed that lung cancer death was frequently identified in deceased patients with the UIP pattern compared with those without either ILD or UIP (5/33 patients [15.2%] vs. 5/103 patients [4.9%]) [46]. In the present study, desaturation with exercise (SpO$_2$ <90%) was observed in more than 80% of RA patients with CPFE, and this parameter was identified as a predictive factor for increased lung cancer-related mortality in RA patients (adjusted HR 2.58 [95% CI 1.40–4.77]). Impaired pulmonary function may contribute to higher mortality risk in lung cancer patients with CPFE compared with those without CPFE. The overall approach to management of lung cancer in CPFE patients is similar to that of other populations [15]. However, increased morbidity and mortality related to lung cancer treatments in CPFE patients were reported to limit various forms of standard-of-care cancer treatment [47], which may, at least in part, explain the poor outcomes in CPFE patients with lung cancer.

In the present study, the predominant histological type was adenocarcinoma in both RA and non-RA groups (52.4% vs. 71.1%). However, a larger proportion of squamous cell carcinoma was observed in RA patients than in non-RA patients (32.9% vs. 15.8%). Similar findings were reported in several single-center cohort studies with RA patients [45, 48]. Recent studies have shown that IPF is a condition predisposing to the development of multiple types of lung cancer and that squamous cell carcinoma is the most common histological type in the setting of IPF or idiopathic CPFE, followed by adenocarcinoma [15, 16, 18, 21]. The higher proportion of squamous cell carcinoma in RA patients may be explained by the higher prevalence of CPFE in RA patients compared with non-RA patients.

In the present study, CPFE was more frequently observed in RA patients than in non-RA patients at the time of lung cancer diagnosis. Considering the possible contribution of idiopathic CPFE to the risk of lung cancer development [22], the higher prevalence of CPFE in RA patients may explain the increased risk of lung cancer in RA patients compared with the general population. Mass screening of all RA patients using HRCT is not supported by current evidence [26], but thorough clinical evaluation and regular screening for lung cancer using imaging are needed in RA patients with CPFE to enable early diagnosis of lung cancer and better prognosis. We also found that RA patients with CPFE had significantly increased lung cancer-related mortality compared with those without ILD or emphysema. The risk of acute exacerbation of ILD is of particular concern in patients with CPFE after surgical resection, radiation, and chemotherapy [15]. The limited choices of lung cancer treatment and

treatment-associated complications may contribute to the increased lung cancer-related mortality in CPFE patients. It is necessary to evaluate the surgical tolerance and to closely monitor pulmonary conditions during lung cancer treatment in RA patients with CPFE.

There are some limitations to this study. First, we did not include the type and implementation of cancer treatment used during follow-up into survival analysis as predictor variables because they were time-varying covariates. Instead, we adopted surgery performed within 1 month after cancer diagnosis for survival analysis. Second, this was a retrospective cohort study, which may confer certain inherent limitations, such as quality of recorded information and/or reliance on the memory of subjects. However, the electronic medical records in the databases of participating institutions allowed us to obtain accurate demographic and clinical data at the time of lung cancer diagnosis, as well as survival status and causes of death. Considering that one-fourth of participants were lost to follow-up, we used the CIF and Fine-Gray competing risks regression methods as survival analysis. Third, we could not examine the effect of DMARD use on lung cancer-related mortality because DMARDs were discontinued during follow-up periods after cancer diagnosis. Given the role of the immune system in tumor surveillance, these drugs could increase the risk of tumor recurrence [38]. Fourth, lung biopsy samples from ILD lesions were not available. Lung biopsy is the gold standard for the classification of UIP patterns. In many cases, however, treating physicians did not obtain biopsy samples from non-cancerous lesions because of concern about the increased risk of biopsy-related complications. The risk of a lung biopsy could outweigh the benefit of establishing a secure diagnosis of UIP. Fifth, the prognostic utility of plasma biomarkers was not evaluated in this study. Further studies are needed to establish which biomarkers could be used to predict outcomes in RA patients with lung cancer and CPFE. Finally, this study was conducted in five community hospitals located in the Kyushu region of Japan. Therefore, our results may not be generalizable to other geographical areas.

## Conclusions

In the present study, we showed a higher prevalence of CPFE at the time of new diagnosis of lung cancer in RA patients compared with non-RA patients. The risk of lung cancer-related death was 2.5 times higher in RA patients than in non-RA patients and was doubled in patients with CPFE compared with patients without pulmonary fibrosis or emphysema. The present study supports the idea that close monitoring and optimal treatment strategies tailored to patients with RA and CPFE are important to improve the poor prognosis of lung cancer in this patient population. The definitions and diagnostic criteria of CPFE as a syndrome have recently been proposed but are not yet widely recognized. Increased awareness of the effect of CPFE on lung cancer risk and mortality is essential for physicians responsible for treating RA patients. We also need to evaluate whether the progression or severity of CPFE may be associated with lung cancer risk and mortality in patients with RA. Further prospective studies involving larger sample sizes of RA patients with lung cancer are warranted to validate our results.

## Supporting information

**S1 Checklist. STROBE statement—checklist of items that should be included in reports of observational studies.**
(DOCX)

**S1 Fig. HRCT scans of a 65-year-old man with rheumatoid arthritis and CPFE.** HRCT images show the coexistence of emphysema (A) and pulmonary fibrosis (B, the UIP pattern of fibrotic ILD). The images show a typical distribution of the disease in CPFE. The zones of

fibrosis and emphysema are completely separated. Paraseptal emphysema is localized to the upper lobes and fibrosis characterized by honeycombing and traction bronchiectasis is localized to the lung bases. Reticular abnormality is also present. HRCT, high-resolution computed tomography; RA, rheumatoid arthritis; CPFE, combined pulmonary fibrosis and emphysema; UIP, usual interstitial pneumonia; ILD, interstitial lung disease.
(TIF)

**S1 Table. DMARD therapy at diagnosis of lung cancer.**
(DOCX)

**S2 Table. Comparison of RA-related factors among lung cancer patients grouped according to HRCT-based CPFE diagnosis.**
(DOCX)

**S3 Table. Predictive factors for lung cancer-related death in RA patients newly diagnosed with lung cancer (Model 1).**
(DOCX)

**S4 Table. Predictive factors for lung cancer-related death in RA patients newly diagnosed with lung cancer (Model 2).**
(DOCX)

## Author Contributions

**Conceptualization:** Shunsuke Mori, Yukitaka Ueki, Kazuyoshi Nakamura.

**Data curation:** Shunsuke Mori, Yukitaka Ueki, Kazuyoshi Nakamura, Toshihiko Hidaka, Koji Ishii, Tomoya Miyamura.

**Formal analysis:** Shunsuke Mori, Mizue Hasegawa, Kazuyoshi Nakamura, Kouya Nakashima.

**Funding acquisition:** Shunsuke Mori.

**Investigation:** Shunsuke Mori, Yukitaka Ueki, Kazuyoshi Nakamura, Toshihiko Hidaka, Koji Ishii, Hironori Kobayashi, Tomoya Miyamura.

**Methodology:** Shunsuke Mori, Yukitaka Ueki, Kazuyoshi Nakamura.

**Project administration:** Shunsuke Mori.

**Resources:** Shunsuke Mori, Yukitaka Ueki, Kazuyoshi Nakamura, Toshihiko Hidaka, Koji Ishii, Hironori Kobayashi, Tomoya Miyamura.

**Supervision:** Shunsuke Mori.

**Validation:** Shunsuke Mori, Mizue Hasegawa, Kazuyoshi Nakamura, Kouya Nakashima.

**Visualization:** Shunsuke Mori.

**Writing – original draft:** Shunsuke Mori.

**Writing – review & editing:** Shunsuke Mori, Yukitaka Ueki, Mizue Hasegawa, Kazuyoshi Nakamura, Kouya Nakashima, Toshihiko Hidaka, Koji Ishii, Hironori Kobayashi, Tomoya Miyamura.

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
