## [Decision Letter · Decision Letter 0]

27 Sep 2023

PONE-D-23-17842Impact of combined pulmonary fibrosis and emphysema on lung cancer-related mortality in patients with and without rheumatoid arthritis: a multicenter retrospective cohort studyPLOS ONE

Dear Dr. Mori,

Thank you for submitting your manuscript to PLOS ONE. After careful consideration, we feel that it has merit but does not fully meet PLOS ONE’s publication criteria as it currently stands. Therefore, we invite you to submit a revised version of the manuscript that addresses the points raised during the review process.

We look forward to receiving your revised manuscript.

Kind regards,

Keiko Hosohata, Ph.D.

Academic Editor

PLOS ONE

Journal Requirements:

“This study was supported by research funds from the National Hospital Organization (NHO), Japan. For the promotion of research in medical science, the NHO provides research funds to clinical research centers of affiliated institutions. The allocation of research funds is determined based on previous research achievements of each institution. Hence, there is no available grant/award number. The funder had no role in the study design, data collection and analysis, decision to publish, or manuscript preparation.”

3. Thank you for stating the following in the Competing Interests/Financial Disclosure* section:

“I have read the journal's policy and the authors of this manuscript have the following competing interests: S. Mori received honoraria for lectures from AbbVie GK, Eli Lilly Japan K.K., Pfizer Japan Inc., Chugai Pharmaceutical Co. Ltd., Janssen Pharmaceutical K.K., Boehringer Ingelheim Japan, and Taisho Pharma Co., Ltd. and received research funds from AbbVie GK, Asahikasei Pharma Corp, and Chugai Pharmaceutical Co., Ltd. Y. Ueki received honoraria for lectures from AbbVie GK, Eli Lilly Japan K.K., Pfizer Japan Inc., Asahikasei Pharma Corp., Astellas Pharma Inc., Bristol-Myers K.K., Chugai Pharmaceutical Co. Ltd., Janssen Pharmaceutical K.K., Mitsubishi Tanabe Pharma Co., Ono Pharmaceutical Co., and Takeda Pharmaceutical Co., Ltd. T. Hidaka received honoraria for lectures from AbbVie GK, Eli Lilly Japan K.K., Pfizer Japan Inc., Asahi Kasei Pharma Corp., Bristol-Myers K.K., Chugai Pharmaceutical Co., Ltd., and Eisai Co. K. Nakamura received honoraria for lectures from AstraZeneca K.K. The other authors had no financial relationships that could create a potential conflict of interest or the appearance of a conflict of interest with regard to the work.”

We note that one or more of the authors are employed by a commercial company

Reviewers' comments:

Reviewer's Responses to Questions

**Comments to the Author**

1. Is the manuscript technically sound, and do the data support the conclusions?

Reviewer #1: Yes

Reviewer #2: Yes

2. Has the statistical analysis been performed appropriately and rigorously? 

Reviewer #1: Yes

Reviewer #2: Yes

3. Have the authors made all data underlying the findings in their manuscript fully available?

Reviewer #1: Yes

Reviewer #2: Yes

4. Is the manuscript presented in an intelligible fashion and written in standard English?

Reviewer #1: Yes

Reviewer #2: Yes

5. Review Comments to the Author

Reviewer #1: The study is a clinical investigation of the association between fibrosis, enphysema and lung cancer. The study was conducted properly: All statistical analyses were performed correctly, and the data were clearly described. However, the data lack originality and do not improve the current clinical management of lung cancer. The study presented is merely descriptive and the authors should go deeper to provide a novel insight into the role of CPFE in lung cancer.

Here, I will provide some comments and suggestions to improve the manuscript and add clinical value, in my humble opinion.

1) The authors should enrich the presented data with exemplary HRCT images for CPFE and ILD to better describe the presented diagnostic guidelines.

2) The authors should also include H&E images with pathological annotations. It is likely that most of these patients had at least one biopsy. In my opinion, it would be interesting to evaluate the number of fibroblastic foci and the expression levels of some IHC markers such as aSMA and collagen 1 to read out CPFE. Can there be an association between foci and collagen deposition and survival?

3) Can the authors extend the analysis to plasma markers? This would improve the manuscript immensely.

4) The author has excluded treatments as covariates. However, fibrosis is known to influence response to drugs, especially immunotherapy. It would be interesting to me to at least check the association between CPFE and drug response.

5) In the discussion, please indicate how the findings can improve the current management of patients with CPFE-associated lung cancer. Can they be treated differently? Can surgery be used earlier? ... and please add any other suggestions.

Minor comments:

1) Sample size: please provide a more "statistically" correct explanation. The description given is not acceptable.

Reviewer #2: The authors have presented a well written manuscript. Findings from the study have been reported in an intelligible manner. Authors have providing an interesting comparison between RA and non RA patients with lung cancer and CPFE. The statistical analyses were rigorously performed. Additionally it will be interesting using the data from the study for the authors to compare the duration of RA and CPFE and disease progression and mortality for lung cancer in these patients.

6. PLOS authors have the option to publish the peer review history of their article (what does this mean?). If published, this will include your full peer review and any attached files.

Reviewer #1: **Yes: **Antonio Agostini PhD

Reviewer #2: No

---

## [Author Response · Author response to Decision Letter 0]

27 Oct 2023

Response to Reviewers

We are most grateful to the reviewers for their valuable comments as well as the time and energy spent for our manuscript. We have made the requested changes and added new information to the manuscript in response to their insightful comments. All alterations are highlighted in red text in the revised manuscript. We are confident that the manuscript has benefited from the reviewers’ useful comments and suggestions.

Below are point-by-point replies to the reviewers’ comments.

Reply to Reviewer 1

We wish to express our appreciation to the reviewer for the comment regarding our study. We understand that it is important to provide novel insights into the role of combined pulmonary fibrosis and emphysema (CPFE) in lung cancer pathogenesis. However, we must emphasize that this is the first cohort study to examine the impact of CPFE on lung cancer risk and lung cancer-related mortality in patients with rheumatoid arthritis (RA). RA is a chronic inflammatory disease, and the high prevalence of interstitial lung disease (ILD) in RA is well known. Additionally, the higher risk of lung cancer in RA patients compared with that in the general population has been reported previously. Recently, several risk factors for lung cancer in RA, such as male sex, smoking, and RA-associated ILD, have been proposed. RA-associated CPFE should be considered an entity distinct from RA-associated ILD. Nevertheless, few clinical studies have examined possible associations between RA-associated CPFE and lung cancer risk and mortality. First, we need to obtain clinical evidence for these associations among RA patients, then we can move forward to address the underlying molecular mechanisms in the CPFE-associated pathogenesis of lung cancer in RA. In 2022, the research definition and classification criteria of CPFE as a clinical syndrome were formalized in an official statement from the American Thoracic Society (ATS), the European Respiratory Society (ERS), Japanese Respiratory Society (JRS), and the Latin American Thorax Association (ALAT). Using this new definition of CPFE, we conducted the present multicenter cohort study. To clarify these points, we have revised the title, Abstract, Introduction, and Discussion sections (lines 2, 3, 30, 33–36, 76–99, and 364–366). New references were added in the revised version (Refs. 23, 26, and 28).

1. We thank the reviewer for the comment regarding high-resolution computed tomography (HRCT) images of CPFE and ILD. In response, we have included figures showing pulmonary HRCT scans of emphysema and ILD (UIP pattern) in an RA patient who was diagnosed with CPFE. The images show the typical distribution of the disease in CPFE. Paraseptal emphysema is localized to the upper lobes. Fibrosis, characterized by honeycombing and traction bronchiectasis, is localized to the lower lobes in a background of reticulation. These findings are consistent with the official ATS/ERS/JRS/ALAT clinical practice guideline. We included these HRCT images as S1 Fig in the revised manuscript (S1 Fig; lines 256–257 and 499–508). 

2. We appreciate the reviewer’s comment regarding biopsy data from CPFE and ILD. This cohort study was performed retrospectively in real-world clinical settings for RA and non-RA patients who were diagnosed with lung cancer between June 2006 and December 2021. Pulmonary HRCT images and histological data of lung cancer were obtained from all participants. Unfortunately, lung biopsy samples from fibrosis lesions were available for only a few patients with CPFE or ILD. We understand that lung biopsy is the gold standard for the classification of UIP patterns. For patients with HRCT findings suggestive of a diagnosis other than IPF, histological examinations might have been useful to exclude other known causes of pulmonary fibrosis. In many cases, however, the treating physicians did not perform lung biopsy because of concerns about the risk of biopsy-related complications. The risk of lung biopsy could outweigh the benefit of establishing a firm diagnosis of the UIP. For patients with RA-associated ILD, histological confirmation was not required because RA was definitively diagnosed as the cause of ILD. In addition, the utility of detailed histopathological scoring systems in day-to-day clinical management of patients with pulmonary fibrosis has not been evaluated. For this study, we collected HRCT images taken at the time of lung cancer diagnosis from all participants, which were viewed in random order and independently by three experts (a board-certified radiologist and two board-certified pulmonologists). We are confident with the HRCT-based diagnoses in this study. 

We agree with the reviewer’s suggestion that it would be valuable to examine the number of fibroblast foci and immunohistochemistry biomarkers, as well as their associations with lung cancer-related mortality. Regrettably, however, for the above-mentioned reasons, we were unable to do such histological analyses for pulmonary fibrosis in ILD or CPFE patients. Needless to say, at the time of lung cancer diagnosis, treating physicians performed lung biopsies for all patients and obtained small tumor samples, because pathological classification of lung cancer is essential for optimal therapeutic strategies. However, lung biopsy was not performed for non-tumor lesions at that time for ethical reasons based on the above-mentioned risks and benefits. 

We included this lack of histopathological data in the Discussion section as one of the limitations of this study in the revised manuscript (lines 425–430).

3. We are grateful for the reviewer’s suggestion regarding plasma markers. We understand that there are various plasma biomarkers for lung cancer. However, their sensitivity and specificity for lung cancer diagnosis are not sufficiently high for routine clinical use. Moreover, a uniform serum biomarker composition capable of distinguishing lung cancer types has yet to be discovered. Therefore, the treating physicians at our institutions did not measure levels of plasma biomarkers at the time of lung cancer diagnosis. As the reviewer suggested, it would be interesting to evaluate the prognostic utility of plasma biomarkers for lung cancer mortality. However, it was beyond the scope of the present study. We described the need for further studies to address which biomarkers could be used to predict outcomes in RA patients with lung cancer and CPFE in the Discussion section (lines 430–432). 

4. We thank the reviewer for the suggestion to examine the association between CPFE and drug response. Regarding cancer treatment, as we described in the Discussion section as a limitation, we did not include the type and implementation of cancer treatment used during follow-up into survival analysis as predictor variables because they were time-varying covariates. Instead, we adopted surgery performed within 1 month after cancer diagnosis for survival analysis (lines 412–415). Non-surgical cancer treatment included chemotherapy and radiation therapy, which are well known to exacerbate pulmonary fibrosis in ILD and CPFE. Treating physicians did not administer chemotherapy in the majority of ILD or CPFE patients due to concerns about this complication. Accordingly, we were unable to examine the effect of ILD or CPFE on response to anticancer drugs. We added this information to the Results section (lines 269–270). 

Regarding disease-modifying antirheumatic drugs (DMARDs), as we listed in S1 Table, a number of DMARDs were used before the diagnosis of lung cancer (methotrexate, biological DMARDs, Janus kinase inhibitors, salazosulfapyridine, bucillamine, mizoribine, cyclosporin, and tacrolimus) as well as prednisolone. Given the role of the immune system in tumor surveillance, immunosuppressive or immunomodulatory therapies for RA could be implicated in the risk of malignancy. Therefore, DMARD use at the time of lung cancer diagnosis was not included in the survival analysis because these drugs were discontinued during cancer treatment due to concerns about tumor recurrence. We address this point in the Discussion section as a limitation (lines 422–425). A new reference was added in the revised version (Ref. 38).

Please understand that this was a retrospective observational study in real-world clinical settings. We could not use therapeutic strategies that were not considered beneficial to patients. 

5. We agree with the reviewer’s suggestion to discuss how our findings can improve the current management of lung cancer in RA patients with CPFE. In the present study, CPFE was more frequently observed in RA patients than in non-RA patients at the time of lung cancer diagnosis. Considering a possible contribution of idiopathic CPFE to the risk of lung cancer development, the higher prevalence of CPFE in RA patients may, at least in part, explain the increased risk of lung cancer in RA patients compared with the general population. Mass screening of all RA patients using HRCT is not supported by current evidence, but thorough clinical evaluation and regular screening for lung cancer with imaging are needed in RA patients with CPFE, which should enable early diagnosis of lung cancer and better prognosis. We also found that RA patients with CPFE had significantly increased cancer-related mortality compared with those without ILD or emphysema. The risk of acute exacerbation of ILD is of particular concern in CPFE patients after surgical resection, radiation, and chemotherapy. Limited options for lung cancer treatment and treatment-associated complications may explain the increased lung cancer-related mortality in this patient population. It is necessary to evaluate surgical tolerance and to closely monitor pulmonary conditions during lung cancer treatment in RA patients with CPFE. To clarify these points, we added one paragraph in the Discussion section of the revised version (lines 397–411). In addition, several sentences were added to the Abstract and Discussion (lines 58–60, 384–386, and 441-443).

(Minor comment)

We express our appreciation for the reviewer’s comment on sample size. To estimate sample size in a survival study for time-to-event, we need to input the expected probability of event-free survival at the time point of interest. The main aim of this study was to evaluate lung cancer-related death outcomes in RA patients with or without CPFE. We tried to determine an expected probability of lung cancer-related death or probability of event-free survival based on previous data. As we pointed out in the Introduction, however, there is a lack of studies to examine survival probability in RA patients with lung cancer with or without CPFE. Accordingly, it was not feasible to determine the expected probability in these patient groups. We added an explanation that we could not estimate sample size for this study due to lack of previous data to the Patients and Methods section (lines 195–199). In such a case, the STROBE statement for observational studies in epidemiology gives an example to explain how the sample size was arrived at. We followed this guideline (lines 199–201). We hope our cancer-related mortality data in RA patients with CPFE will be helpful to calculate sample size for future studies.

Reply to Reviewer 2

We greatly appreciate the reviewer’s comments about our study overall and the suggestion to compare the duration of RA and CPFE and disease progression and lung cancer-related mortality using the data obtained from this study. To examine the effect of RA duration/progression at baseline on lung-cancer related death outcome, we included these factors into the Fine-Gray competing risks regression analysis. We found that neither RA duration nor Steinbrocker radiological stage (as a measure of RA progression) was associated with lung cancer-related mortality (S3 Table). We also examined the association between RA duration/progression and CPFE complication, and found no significant differences in either RA-related factor among patients with CPFE, those with ILD or emphysema alone, and those without these complications (S2 Table). We included these new data in the Patients and Methods and Results sections of the revised manuscript (lines 226–228, 254–256, and 325–330). New S2 and S3 Tables were also included (lines 493–494 and 496–497). In the present study, we made a definitive diagnosis of CPFE using HRCT scans taken at the diagnosis of lung cancer according to the 2022 official statement of the ATS, ERS, JRS, and ALAT. We were not certain whether the treating physicians had definitively diagnosed these patients as having CPFE before enrollment of our study. Further, disease progression/severity staging of CPFE has not been established at present. Accordingly, we could not determine the duration or progression of CPFE at the time of enrollment of these patients. The definitions and diagnostic criteria of CPFE have recently been proposed by the ATS, ERS, JRS, and ALAT. The statement first proposed to identify CPFE as a syndrome. We hope our study will help physicians increase awareness of the association of CPFE with lung cancer risk and mortality in RA patients. We added these points in the Conclusions section of the revised version (lines 443–448).

---

## [Decision Letter · Decision Letter 1]

3 Jan 2024

PONE-D-23-17842R1Impact of combined pulmonary fibrosis and emphysema on lung cancer risk and mortality in rheumatoid arthritis: A multicenter retrospective cohort studyPLOS ONE

Dear Dr. Mori,

Thank you for submitting your manuscript to PLOS ONE. After careful consideration, we feel that it has merit but does not fully meet PLOS ONE’s publication criteria as it currently stands. Therefore, we invite you to submit a revised version of the manuscript that addresses the points raised during the review process.

We look forward to receiving your revised manuscript.

Kind regards,

Masataka Kuwana, MD, PhD

Academic Editor

PLOS ONE

Journal Requirements:

**Additional Editor Comments:**

The points raised by reviewers are almost completely responded in the revised version. Additional reviewer raised some minor points, which should be useful to improve scientific quality of the manuscript.  

Reviewers' comments:

Reviewer's Responses to Questions

**Comments to the Author**

1. If the authors have adequately addressed your comments raised in a previous round of review and you feel that this manuscript is now acceptable for publication, you may indicate that here to bypass the “Comments to the Author” section, enter your conflict of interest statement in the “Confidential to Editor” section, and submit your "Accept" recommendation.

Reviewer #1: (No Response)

Reviewer #2: All comments have been addressed

Reviewer #3: (No Response)

2. Is the manuscript technically sound, and do the data support the conclusions?

Reviewer #1: Partly

Reviewer #2: Yes

Reviewer #3: Yes

3. Has the statistical analysis been performed appropriately and rigorously? 

Reviewer #1: Yes

Reviewer #2: Yes

Reviewer #3: Yes

4. Have the authors made all data underlying the findings in their manuscript fully available?

Reviewer #1: No

Reviewer #2: Yes

Reviewer #3: Yes

5. Is the manuscript presented in an intelligible fashion and written in standard English?

Reviewer #1: Yes

Reviewer #2: Yes

Reviewer #3: Yes

6. Review Comments to the Author

Reviewer #1: I still think that this sudy present several limitattions, but I understand that in the lack of material there is nothing more that the authors could do. I gladly accepted rhe fact that authors discolsed all the limitations in the manuscript.

Hope that this study is a starting point for future perspectives.

Reviewer #2: (No Response)

Reviewer #3: The authors aimed to address the effect of CPFE on risk for lung cancer and lung-cancer related mortality in patients with RA. CPFE have not received much attention in patients with RA and this paper can add insights on the subject.

1. RA itself is a well-known risk factor malignancy including lung cancer. It can be a significant confounding factor when comparing incidence of lung cancer in RA and non-RA patients. Comparison with RA patients with or without CPFE should be made.

2. RA disease activity can have significant impact on mortality. Was it considered in the analysis?

3. Pulmonary arterial hypertension is commonly associated with CPFE and impact mortality. What were the incidences of PAH and did it show association with mortality?

4. Infection is a complication that is more common in patients with RA. Was death by infection included cancer-related mortality or categorized separately?

5. How was the lung function in patients with CPFE. It can also be a significant factor influencing mortality and should be included in the analysis

7. PLOS authors have the option to publish the peer review history of their article (what does this mean?). If published, this will include your full peer review and any attached files.

Reviewer #1: **Yes: **Antonio Agostini, PhD

Reviewer #2: No

Reviewer #3: No

---

## [Author Response · Author response to Decision Letter 1]

20 Jan 2024

Response to Reviewers

We are most grateful to the editor and reviewers for their valuable comments and for the time and energy spent reviewing our manuscript. We have made the requested changes and added new information to the manuscript in response to their insightful comments. All alterations are highlighted in red text in the revised manuscript. We are confident that the manuscript has benefited from the reviewers’ useful comments and suggestions.

Below are point-by-point replies to the reviewers’ comments.

Reply to Reviewer 1

We appreciate the reviewer’s understanding that there were several inevitable limitations of our study because this was a retrospective observational study in a real-world clinical setting. Despite these limitations, we obtained clinical evidence for the impact of combined pulmonary fibrosis and emphysema (CPFE) on lung cancer risk and lung cancer-related mortality in patients with rheumatoid arthritis (RA). As this reviewer kindly suggested, we hope that our research will stimulate further investigation of underlying molecular mechanisms in the CPFE-associated pathogenesis of lung cancer in RA.

Reply to Reviewer 3

1. We thank the reviewer for the suggestion to compare lung cancer mortality data between RA patients with CPFE and those without this complication because RA per se is a well-known risk factor for lung cancer. The subjects of this study were patients with and without RA who had been newly diagnosed with lung cancer in participating institutions. We performed a retrospective follow-up study of these patients until lung cancer-related death, other-cause death, or the end of the study. In the original manuscript, we compared the crude incidence rate of lung cancer-related death as well as the death probability estimated by the cumulative incidence function (CIF) for the three patient groups (CPFE, interstitial lung disease [ILD] or emphysema alone, and the group without these conditions) in all patients as well as separately in RA patients and non-RA patients. The crude incidence rate was highest in the CPFE group in all patients, and the same differences were observed separately in RA and non-RA patients. These data are shown in Table 3 of the manuscript, and we describe these comparison data in the Results section (lines 307–313) of the revised manuscript. Additionally, the CIF mortality estimates over time were significantly worse in the CPFE group. The significant differences among the three patient groups were observed in all patients (p<0.001 with Gray’s test), as well as separately in RA and non-RA patients (p=0.006 in RA patients and p<0.001 in non-RA patients). Please see the Results section (lines 314–322) and Table 3. In the revised manuscript, we added new comparison data of the CIF mortality estimates between RA and non-RA patients grouped according to the HRCT-based CPFE diagnosis. RA patients were estimated to be at higher risk of lung cancer-related death over time compared with non-RA patients in each of the three patient groups (p<0.001 with Gray’s test). These comparison data are described in the Results (lines 322–324) and Table 3 footnote. 

2. We thank the reviewer for the suggestion to examine the association between RA disease activity and lung cancer-related mortality. In the present study, RA disease activity was well controlled in 68.3% of patients at lung cancer diagnosis (i.e., patients had low disease activity or remission status). There was no significant difference in RA disease activity among the CPFE group, the ILD or emphysema alone group, and the group without these complications. We included this information in the Results section (lines 246 and 261–263) as well as Table 1 and S2 Table of the revised manuscript. To examine the effect of RA disease activity at baseline on lung cancer-related death, we included this variable in the Fine-Gray competing risks regression analysis for RA patients. We found that RA disease activity was not associated with lung cancer-related mortality. We added this information to the Results section (lines 343–348), the Discussion section (lines 380 and 381) as well as in S3 Table. 

3. We appreciate the reviewer’s comment regarding the incidence of pulmonary arterial hypertension (PAH). Among patients diagnosed as having CPFE based on HRCT scans taken at the diagnosis of lung cancer, we found 2 patients with PAH. This diagnosis had been made based on data of echocardiography and right heart catheterization (1 in an RA patient and 1 in a non-RA patient). Both patients died during follow-up, and these cases were categorized as lung cancer-related death (oncological emergency events). This information was included in the Results section of the revised version (lines 258–260, 288, and 289). 

4. We understand the reviewer’s question regarding whether or not death caused by infectious diseases was included in lung cancer-related death. As mentioned in the “Patients and methods” section of the original manuscript, lung cancer-related death was defined as death caused by cachexia, organ failure, infectious disease (such as pneumonia and sepsis), complications of cancer therapies, or oncological emergencies (line 178–182). In our cohort, 36 patients with lung cancer died from infectious diseases (pneumonia and sepsis; 16 in RA patients and 20 in non-RA patients), and we included these cases in the lung cancer-related death category. We added this information to the Results section of the revised manuscript (lines 287 and 288). As shown in Table 2, there was no other-cause death in RA patients. 

5. We are grateful for the reviewer’s suggestion to examine the influence of pulmonary function on mortality in CPFE patients. In the present study, we made a definitive diagnosis of CPFE at study enrollment based on HRCT scans taken at the diagnosis of lung cancer according to the 2022 official guidelines, and we identified 81 CPFE patients (33 in RA patients and 48 in non-RA patients). However, the vast majority of non-RA patients visited the respiratory department of our institution for further examination and treatment of lung cancer. They did not receive pulmonary function tests upon diagnosis of lung cancer. We were not certain whether the general practitioners referring these patients were aware of the presence of CPFE in these non-RA patients or whether they had performed pulmonary function tests for these patients before referral to our respiratory department. Data of percutaneous oxygen saturation [SpO₂] after a 6-minute walk or equivalent, which is an important parameter for severe pulmonary fibrosis, were available in all RA patients. We found that 47.6% of RA patients showed desaturation with exercise, defined as SpO₂ <90% during 6-minute walk. The rates of patients with desaturation with exercise were significantly different among the CPFE group, the ILD or emphysema alone group, and the group without these conditions (p<0.001). We added this information to the “Patients and methods” section (lines 142 and 143), the Results section (lines 246–248 and 264 –265) as well as in Table 1 and S2 Table. 

To examine the effect of desaturation with exercise at baseline on lung-cancer related death outcomes, we included this predictor variable instead of HRCT-based CPFE diagnosis in the multivariable Fine-Gray competing risks regression analysis. HRCT-based CPFE diagnosis was not included in the multivariable analysis with desaturation with exercise because both predictor variables were highly correlated. We found that, like HRCT-based diagnosis, desaturation with exercise was an independent predictive factor for lung cancer-related mortality (adjusted HR 2.58 [95% CI 1.40–4.77]). We included these new data in the Results section (lines 348–354) and the Discussion section (lines 405–410) of the revised manuscript. A new supplementary table was added to the revised version (S4 Table; lines 527 and 528).

---

## [Decision Letter · Decision Letter 2]

29 Jan 2024

Impact of combined pulmonary fibrosis and emphysema on lung cancer risk and mortality in rheumatoid arthritis: A multicenter retrospective cohort study

PONE-D-23-17842R2

Dear Dr. Mori,

We’re pleased to inform you that your manuscript has been judged scientifically suitable for publication and will be formally accepted for publication once it meets all outstanding technical requirements.

Kind regards,

Masataka Kuwana, MD, PhD

Academic Editor

PLOS ONE

Additional Editor Comments (optional):

Reviewers' comments:

Reviewer's Responses to Questions

**Comments to the Author**

1. If the authors have adequately addressed your comments raised in a previous round of review and you feel that this manuscript is now acceptable for publication, you may indicate that here to bypass the “Comments to the Author” section, enter your conflict of interest statement in the “Confidential to Editor” section, and submit your "Accept" recommendation.

Reviewer #3: All comments have been addressed

2. Is the manuscript technically sound, and do the data support the conclusions?

Reviewer #3: Yes

3. Has the statistical analysis been performed appropriately and rigorously? 

Reviewer #3: Yes

4. Have the authors made all data underlying the findings in their manuscript fully available?

Reviewer #3: Yes

5. Is the manuscript presented in an intelligible fashion and written in standard English?

Reviewer #3: Yes

6. Review Comments to the Author

Reviewer #3: Thank you for answering all questions. The question raised were all addressed. No additional comments.

7. PLOS authors have the option to publish the peer review history of their article (what does this mean?). If published, this will include your full peer review and any attached files.

Reviewer #3: No

---

## [Editor Report · Acceptance letter]

17 Feb 2024

PONE-D-23-17842R2 

PLOS ONE

Dear Dr. Mori, 

I'm pleased to inform you that your manuscript has been deemed suitable for publication in PLOS ONE. Congratulations! Your manuscript is now being handed over to our production team.

Kind regards, 

on behalf of

Prof. Masataka Kuwana 

Academic Editor

PLOS ONE